# Meaning in life, positive cognition, and learning motivation: A mediational analysis among Chinese college students

**Shi-hong Liu[1]☯, Xiao-li Gong[2]☯, Qi Ji[1]‡, Qiao-qiao Li[1]‡, Ming-hui Kong[1,3]‡, Xiao-hui Du[1]‡, Lei-ying Xiang[1,3]‡, Zhi-ru Zhu ![ORCID][1]\***

**1** Department of Medical Psychology, Army Medical University, Chongqing, China, **2** High Altitude Medicine Department of Army 953 Hospital, Xizang, China, **3** School of Educational Science, Chongqing Normal University, Chongqing, China

☯ These authors contributed equally to this work and are co-first authors.
‡ These authors also contributed equally to this work.
\* zhuzr2013@163.com

## Abstract

### Background

While a sense of meaning in life is known to boost students' learning motivation, the psychological process explaining *how* this happens is not well understood. This study investigated whether positive thinking (positive cognition) acts as the crucial bridge connecting these two.

### Methods

We conducted a survey with 345 university students in Chongqing, China. They completed questionnaires measuring their sense of meaning in life (both the search for meaning and the feeling of having it), their tendency for positive thinking, and their motivation for learning (both for internal satisfaction and external rewards).

### Results

We found a clear link: students with a stronger sense of life meaning also reported higher levels of positive thinking and greater learning motivation. More importantly, positive thinking was a key pathway. A students' sense of life meaning appeared to first nurture a more positive mindset, and this positive mindset, in turn, was what fueled their motivation to learn. This effect was particularly strong for intrinsic motivation (learning for the love of it). For extrinsic motivation (learning for rewards), positive thinking was also a vital link, sometimes serving as the primary channel through which life meaning influenced motivation.

### Conclusion

Positive thinking is a key ingredient that transforms a sense of life meaning into tangible motivation for learning. Our findings suggest that universities could better support

**Data availability statement:** Informed consent was obtained from all participants prior to their participation. Research data supporting this publication are available at figshare. org(DOI:10.6084/m9.figshare.29504990).

**Funding:** This work was supported by grants from the Army Medical University (Nos. Y2025W14 & 2022B12) received by Zhi-ru Zhu.

**Competing interests:** NO authors have competing interests.

students by not only helping them explore life's meaning but also by actively fostering positive cognitive skills. Such programs could improve students' academic engagement and overall well-being.

---

## 1. Introduction

Amidst the rapid national development and economic advancement in China, there has been a notable increase in material standards of living. However, commensurate improvements in subjective well-being have not been uniformly observed, and complex trends, including potential stagnation or decline, have been noted within certain population segments. Intensified social competition and rapid societal transformations may contribute to existential questioning among some individuals, particularly university students [1]. Historically, while emphasizing academic achievement, the educational system in China may have devoted insufficient attention to the cultivation of students' sense of meaning in life (MiL) and their holistic psychological development. Recent occurrences of incidents related to student mental health underscore the imperative of fostering healthy perspectives on life and values, alongside promoting positive self-perception, during formative education [2,3].

A substantial body of recent research confirms significant positive correlations between MiL (or its core components, such as a sense of purpose) and university students' learning engagement, academic motivation, and academic achievement [1,4,5]. Possessing a clear sense of life can provide an intrinsic drive for students' learning; understanding the value and objectives of their studies helps stimulate academic enthusiasm [4].

Despite this established link, a critical gap remains in the literature: the underlying psychological mechanisms that explain how a sense of life meaning translates into enhanced learning motivation are not yet well understood. While some studies have explored mediators such as self-efficacy [5] or academic emotions [6,7], the role of core cognitive processes, such as an individual's fundamental style of thinking, has been largely overlooked. To address this gap, the present study proposes and tests Positive Cognition—an individual's habitual tendency to interpret experiences optimistically and constructively—as a key mediating factor. Elucidating this mediating mechanism is crucial, as it can provide targeted empirical evidence for developing novel educational interventions aimed at enhancing university students' learning motivation.

Therefore, the present study aimed to investigate the mediating role of positive cognition in the relationship between meaning in life and learning motivation among Chinese university students.

## 2. Literature review

### 2.1. Theoretical framework construction

The primary challenge in understanding the link between students' existential perceptions and their academic drive lies in identifying the underlying psychological

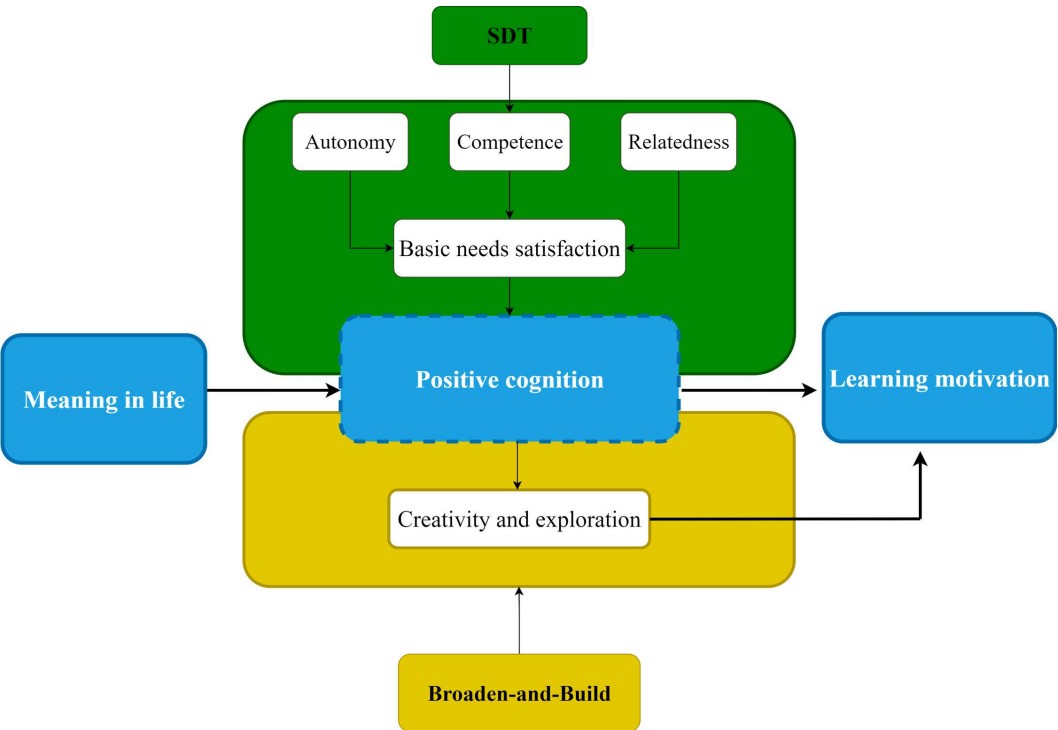

**Fig 1. Theoretical model diagram.**

mechanisms. While Self-Determination Theory (SDT) [8–10] provides a foundational framework for motivation, it does not fully elaborate on the specific cognitive processes that translate broad life purpose into specific academic behaviors. SDT posits that the satisfaction of innate needs for autonomy, competence, and relatedness fuels well-being and intrinsic motivation [9,11], but the cognitive "engine" of this process requires further exploration [12].

To address this limitation, this study integrates SDT with Fredrickson's Broaden-and-Build Theory [13]. The Broaden-and-Build Theory complements SDT by explaining how positive psychological states function. It suggests that positive cognitions and emotions broaden an individual's thought-action repertoires, fostering creativity and exploration, which in turn builds durable personal resources, including sustained motivation.

Therefore, by integrating SDT and the Broaden-and-Build Theory, we construct a more comprehensive model. This model posits that a strong sense of meaning in life contributes to the satisfaction of basic psychological needs (as per SDT), which fosters a positive cognitive style. This positive cognition then acts as the mechanism that broadens students' engagement with learning and builds their long-term learning motivation (as per Broaden-and-Build Theory). This integrated framework allows us to move beyond a simple correlational understanding to a more nuanced, process-oriented explanation. The proposed theoretical model is illustrated in S1 Fig.

## 2.2. The role meaning in life

Meaning in Life (MiL) is generally conceptualized as an individual's subjective perception of the purpose, significance, and coherence of their existence [14,15]. It is considered a critical component of psychological well-being and positive adaptation [16]. Early existential philosophers addressed the individual construction of meaning in life. Contemporary psychological research, particularly within positive psychology, identifies 'life goals' or 'purpose' as a core constituent of

MiL, providing substantive content to the sense of meaning. Researchers commonly distinguish between two dimensions of MiL: the presence of meaning, reflecting the degree to which individuals currently perceive their lives as meaningful, and the search for meaning, denoting the active process and motivation to find or establish life meaning [17]. This distinction remains salient in the current research, as the presence of meaning is typically associated with enhanced well-being and adjustment, whereas the implications of the search for meaning appear more complex [17,18]. A strong presence of meaning is foundational, as it provides students with a "why" for their efforts, which, according to SDT, is a key element in fostering a sense of autonomy and purpose.

## 2.3. The role of learning motivation

Learning motivation, the internal force propelling student learning behaviour, encompasses intrinsic motivation, directed towards the inherent enjoyment and satisfaction of the learning activity itself, and extrinsic motivation, driven by external rewards, punishments, or pressures [10,19,20]. From an SDT perspective, intrinsic motivation represents the prototype of autonomous, self-determined behavior [21,22]. When students' needs for autonomy, competence, and relatedness are met within the learning environment, their intrinsic motivation is naturally enhanced [23]. Furthermore, such self-determined motivation is a robust predictor of positive academic outcomes like achievement and academic buoyancy [24]. In the contemporary context, advocating for lifelong and autonomous learning and stimulating and sustaining university students' learning motivation, particularly intrinsic motivation, is critically important [6,25,26]. Thus, understanding how to cultivate this intrinsic drive is a central goal for educational practice.

## 2.4. The mediating role of positive cognition

The central proposition of this study is that positive cognition serves as a critical bridge linking meaning in life to learning motivation. We selected positive cognition as the mediator for several theoretical reasons. First, from a cognitive appraisal perspective, meaning in life is not merely an abstract feeling but is fundamentally shaped by how individuals interpret their experiences [27]. A strong sense of purpose provides a stable cognitive framework that encourages individuals to view challenges as manageable and opportunities as abundant, which is the very essence of positive cognition. Within the SDT framework, this connection can be further elucidated. When individuals perceive their lives as meaningful, they are more likely to feel a sense of autonomy in their choices and competence in pursuing their life goals. This satisfaction of basic psychological needs naturally leads to more positive cognitive and emotional states [11]. This is where the Broaden-and-Build Theory becomes particularly relevant. These positive cognitive states are not passive byproducts; they actively work to broaden students' thinking [13]. For instance, a student with a positive cognitive style may see a difficult assignment not as a threat, but as an opportunity to learn and grow, thereby expanding their repertoire of academic coping strategies. Positive Cognition, defined herein as an individual's habitual tendency to interpret experiences, evaluate the self, and anticipate the future in an optimistic, positive, and constructive manner, is thus a strong candidate for explaining the link between MiL and learning motivation. Recent studies have highlighted the role of positive psychological resources (such as psychological capital, encompassing optimism, hope, and self-efficacy) in fostering academic engagement and well-being concepts closely related to positive cognition [4]. Individuals with positive cognitions may more readily discover or construct meaning in life, thereby deriving growth [26]. This positive cognitive style—characterized by focusing on positive information, maintaining hope for the future, and believing in one's capacity to cope with challenges—is crucial for sustaining learning effort, setting, and pursuing academic goals, and consequently may influence learning motivation. The satisfaction of SDT needs, nurtured by a sense of life meaning, provides the psychological fuel for maintaining such positive cognitions, which then directly feed into a more self-determined and robust learning motivation [28]. Ultimately, the satisfaction of SDT needs, nurtured by a sense of life meaning, provides the psychological fuel for maintaining such positive cognitions [11,29]. These cognitions, in turn, build lasting intellectual and motivational resources, as predicted

by the Broaden-and-Build Theory, directly feeding into a more self-determined and robust learning motivation. Previous research has also provided preliminary evidence suggesting links between positive cognition or related constructs (e.g., attention to positive information and perspective-taking) and MiL experience [27].

Although constructs such as self-efficacy, resilience, and optimism have also been shown to mediate similar psychological pathways, positive cognition was chosen in this study because it encompasses a broader tendency to interpret experiences optimistically and constructively, which simultaneously includes elements of hope, future orientation, and positive self-appraisal. Furthermore, positive cognition is potentially more malleable through cognitive interventions relevant to educational contexts than more trait-like constructs such as resilience. The validated scale used for positive cognition also operationalizes this construct at the cognitive style level, which is most directly aligned with our goal of understanding cognitive mediational processes.

## 2.5. The current study

In summary, while previous research has laid a solid foundation by linking meaning in life to learning motivation, several gaps necessitate the current investigation. First, there is a lack of research examining the specific mediating role of positive cognition in this relationship. Understanding this cognitive pathway is essential because cognitive patterns are often more malleable and serve as a direct target for psychological interventions. Second, few studies have attempted to integrate these three constructs (meaning in life, positive cognition, and learning motivation) within a coherent theoretical framework like Self-Determination Theory. By doing so, this study moves beyond mere correlation to test a theoretically-driven explanatory model. Finally, given the unique socio-cultural context of contemporary China, exploring these psychological dynamics among university students holds significant practical importance for educational policy and mental health support.

Drawing upon the aforementioned theoretical background and empirical findings, this study posits that meaning in life does not only directly influence learning motivation but also exerts an indirect effect through the mechanism of positive cognition. Specifically, a strong sense of life meaning may foster more positive cognitive habits, which in turn fuel greater learning motivation. Based on the theoretical connections and empirical literature reviewed, we propose the following hypotheses, the hypothesized mediation model is illustrated in S2 Fig.

- Hypothesis 1: MiL will be positively associated with learning motivation.

- Hypothesis 2: MiL will be positively associated with positive cognition.

- Hypothesis 3: Positive cognition will be positively associated with learning motivation.

- Hypothesis 4: Positive cognition will mediate the relationship between MiL and learning motivation.

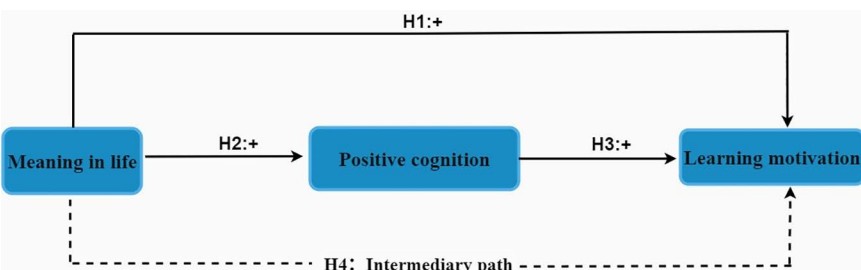

**Fig 2. The hypothesized mediation model.**

## 3. Materials and methods

### 3.1. Participants

A total of 351 college students from a university in Chongqing were selected using random sampling. The recruitment period for this study was from 01/03/2024 to 01/08/2024. All participants were over 18 years of age.After removing missing data and careless responses from the dataset, the final analysis included 345 valid participants, resulting in an efficacy rate of 98.29%. The mean age of subjects was 21.01±2.52 years, with 200 males (21.29±2.47 years) and 145 females (20.64±2.54 years).

The study was approved by the Ethics Committee of the Second Affiliated Hospital of Army Medical University (Approval No. 2023-Yan-127–01, Date: 28 September 2023). Electronic informed consent was obtained from all participants. Prior to starting the online survey, participants were presented with an introductory screen that provided essential information about the study, including its general purpose, the voluntary nature of participation, assurance of strict confidentiality and anonymity of data, and the right to withdraw from the study at any time without penalty. This introductory screen explicitly stated, "If you agree to the above, and have understood the content to be surveyed, please begin".Participants indicated their informed consent by voluntarily proceeding to complete the online questionnaire after acknowledging this statement. This electronic consent procedure was approved by the Ethics Committee. Furthermore, the online questionnaire design included an explicit option for participants to exit the survey at any point before final submission, ensuring their right to withdraw. The survey was conducted anonymously, and data were collected and analyzed in a manner that ensured participants' privacy and confidentiality.

### 3.2. Instruments

**3.2.1. Meaning in life questionnaire (MLQ).** The questionnaire, originally developed by Steger in 2006 and revised by Liu Sisi in 2010 [17,30], includes two aspects: the Presence of Meaning (MLQ-P) and Search for Meaning (MLQ-S). It includes nine items, with items 6, 7, 8, and 9 measuring the Presence of Meaning dimension and items 1, 2, 3,4, and 5 measuring the Search for Meaning dimension. The second item was reverse-scored. The questionnaire was based on a Likert 7-point scale, ranging from "Completely Disagree" (1 point) to "Completely Agree" (7 points). The higher the score, the stronger the meaning of life. In this study, the Cronbach's α coefficient was 0.855 for the entire questionnaire, 0.892 for the Presence of Meaning subscale, and 0.849 for the Search for Meaning subscale.

The fit indices for the MIL Questionnaire were $\chi^2/df = 2.98$, RMSEA = 0.08, CFI = 0.97, NNFI = 0.96, and RMR = 0.12. The Pearson correlation coefficients between other factors were smaller than the square root of the average variance extracted (AVE) values. For the search for meaning subscale, the AVE was 0.59 (> 0.50) and Composite Reliability (CR) was 0.85 (> 0.70), indicating strong convergent validity. For the meaning subscale, the AVE was 0.64 (> 0.50) and CR was 0.90 (> 0.70), also demonstrating good convergent validity (Table 1).

**Table 1. Validity results of variables.**

|  | AVE | CR | Positive Cognition | Seeking Meaning | Possessing Meaning | Intrinsic Motivation | Extrinsic Motivation |
|---|---|---|---|---|---|---|---|
| Positive Cognition | 0.53 | 0.93 | 0.730 |  |  |  |  |
| Seeking Meaning | 0.59 | 0.85 | 0.394 | 0.770 |  |  |  |
| Possessing Meaning | 0.64 | 0.90 | 0.548 | 0.326 | 0.798 |  |  |
| Intrinsic Motivation | 0.51 | 0.89 | 0.632 | 0.384 | 0.632 | 0.715 |  |
| Extrinsic Motivation | 0.28 | 0.72 | 0.498 | 0.303 | 0.224 | 0.398 | 0.526 |

Diagonal numbers represent the square root of AVE.

**3.2.2. Positive cognitive questionnaire.** The Chinese version of the Positive and Negative Information Attention Scale, revised by Feng et al., was adopted [31,32]. This study used the Positive Information Attention subscale.

The subscale comprises 12 items rated on a Likert 5-point scale: "5 (Completely Agree)," "4 (Mostly Agree)," "3 (Uncertain)," "2 (Mostly Disagree)," and "1 (Completely Disagree)." Higher scores reflect a stronger positive cognitive bias. The internal consistency (Cronbach's α) for the positive cognitive subscale was 0.928.

The first-order confirmatory model fit indices for the Positive Cognition Scale were $\chi^2/df = 6.52$, RMSEA = 0.13, CFI = 0.89, NNFI = 0.86, and RMR = 0.04. The Pearson correlation coefficients between the other factors were smaller than the square root of the AVE values. The AVE for positive cognition was 0.53 (> 0.50) and CR was 0.93 (> 0.70), indicating strong convergent validity.

**3.2.3. Learning motivation questionnaire.** Learning motivation is generally categorized into two dimensions: intrinsic and extrinsic [33]. This study used the two-dimensional learning motivation scale designed by Wei, which consists of 16 items divided into intrinsic and extrinsic motivation scales (Cronbach's α coefficients: 0.8023 and 0.8231) [34].

The first-order confirmatory model fit indices for the Learning Motivation Scale were $\chi^2/df = 5.22$, RMSEA = 0.11, CFI = 0.81, NNFI = 0.78, and RMR = 0.09. The Pearson correlation coefficients between the other factors were smaller than the square root of the AVE values. For intrinsic motivation, the AVE was 0.51 (> 0.50) and CR was 0.89 (> 0.70), indicating strong convergent validity. For extrinsic motivation, the AVE was 0.28 and CR was 0.72 (> 0.70).

## 3.3. Statistical analysis

SPSS 24.0 was used for data preprocessing, descriptive analysis, and correlation analysis. The mediating role of positive cognition was investigated by applying the hierarchical regression analysis proposed by Wen. The significance of the mediation effect was verified using the bootstrap method [35].

# 4. Results

## 4.1. Common method bias

Harman's single-factor test was used to assess common method bias. Exploratory factor analysis revealed six factors with eigenvalues greater than one, accounting for a total variance of 62.79%. The first factor accounted for 35.87% of the variance, which was below the critical threshold of 40%, indicating no significant common method bias.

Discriminant Validity: The absolute value of the correlation between potential variables was generally less than the square root of their respective AVE values (Table 1), confirming discriminant validity.

## 4.2. Descriptive statistics a correlation analysis

Descriptive statistics and correlations between meaning in life, learning motivation, and positive cognition are presented in Table 2. The results showed significant positive correlations between life meaning and learning motivation, life meaning and positive cognition, and positive cognition and learning motivation.

## 4.3. Mediation analysis of positive cognition

**4.3.1. Mediation of positive cognition between meaning in life and intrinsic motivation.** Based on Fritz & MacKinnon (2007), Bootstrap mediation analysis requires sample sizes of ≥546 for small effects (e.g., a = 0.14, b = 0.14) and ≥71 for medium effects (e.g., a = 0.39, b = 0.39). With our sample of 345, it exceeds needs for medium effects and

**Table 2. Descriptive statistics and Pearson correlations analysis of variables.**

| | M | SD | Positive Cognition | Seeking Meaning | Possessing Meaning | Intrinsic Motivation | Extrinsic Motivation |
|---|---|---|---|---|---|---|---|
| Positive Cognition | 4.0597 | 0.59546 | | | | | |
| Seeking Meaning | 5.4087 | 1.20832 | 0.394** | | | | |
| Possessing Meaning | 5.2383 | 1.24617 | 0.548** | 0.326** | | | |
| Intrinsic Motivation | 4.0913 | 0.57677 | 0.632** | 0.384** | 0.632** | | |
| Extrinsic Motivation | 3.7225 | 0.51861 | 0.498** | 0.303** | 0.224** | 0.398** | |

**. Correlation is significant at the 0.01 level (2-tailed).

provides ≥80% power for most cases where at least one path (a or b) is medium-sized (≥0.26). Limitations exist if both paths are very small (<0.14). Overall, 345 is adequate for Bootstrap-based mediation in typical social science research with expected path coefficients ≥0.20 [36].

The hierarchical regression analysis method proposed by Zhong-lin Wen was used to examine the mediating role of positive cognition between a sense of meaning in life and intrinsic motivation.

Mediation analyses were conducted in SPSS 24.0 using hierarchical regression and the Bootstrap method via Hayes' process macro (Model 4).The results of the stepwise regression analysis are presented in Table 3, and the bootstrap mediating effect results are shown in Table 4.

From Table 3, it can be seen that in Step 1, the standardized regression coefficient for the independent variable seeking meaning was 0.384, t = 7.693, p < 0.001, and for the independent variable possessing meaning, it was 0.632, t = 15.085 p < 0.001, indicating that both seeking meaning and possessing meaning positively influence intrinsic motivation, validating coefficient c; in Step 2, the standardized regression coefficient for the independent variable seeking meaning was 0.394, t = 7.946, p < 0.001, and for the independent variable possessing meaning, it was 0.548, t = 12.139, p < 0.001, indicating that both seeking meaning and possessing meaning positively influence positive cognition, validating coefficient a; in Step 3, when the independent variable is seeking meaning, the standardized regression coefficient for the mediating variable positive cognition was 0.569, t = 12.721, p < 0.01, when the independent variable is possessing meaning, the standardized regression coefficient for the mediating variable positive cognition was 0.409, t = 9.089, p < 0.01, indicating that positive cognition positively influences intrinsic motivation, validating coefficient b, and showing that positive cognition plays a mediating role between the sense of meaning in life and intrinsic motivation. Meanwhile, the standardized regression

**Table 3. The mediating effect test of positive cognition between sense of meaning in life and intrinsic motivation.**

| Step | Path | Unstandardized Coefficients | | Standardized Coefficients | t | p | Adjusted $R^2$ |
|---|---|---|---|---|---|---|---|
| | X→Y | B | SD | β | | | |
| 1 | Seeking Meaning→Intrinsic Motivation | 0.138*** | 0.024 | 0.384 | 7.693 | 0.000 | 0.145 |
| | Possessing Meaning→Intrinsic Motivation | 0.292*** | 0.019 | 0.632 | 15.085 | 0.000 | 0.397 |
| 2 | Seeking Meaning→Positive Cognition | 0.194*** | 0.024 | 0.394 | 7.946 | 0.000 | 0.153 |
| | Possessing Meaning→Positive Cognition | 0.262*** | 0.022 | 0.548 | 12.139 | 0.000 | 0.298 |
| 3 | Seeking Meaning→Intrinsic Motivation | 0.076*** | 0.021 | 0.159 | 3.553 | 0.000 | 0.418 |
| | Positive Cognition→Intrinsic Motivation | 0.552*** | 0.043 | 0.569 | 12.721 | 0.000 | |
| | Possessing Meaning→Intrinsic Motivation | 0.189*** | 0.021 | 0.407 | 9.055 | 0.000 | 0.513 |
| | Positive Cognition→Intrinsic Motivation | 0.396*** | 0.044 | 0.409 | 9.089 | 0.000 | |

* p < 0.05 ** p < 0.01 ***p < 0.001.

**Table 4. Bootstrap mediated effect test of positive cognition between sense of meaning in life and intrinsic motivation.**

| Path | Effect | Effect Value | 95% CI | | SE | Conclusion |
|---|---|---|---|---|---|---|
| | | | LLCI | ULCI | | |
| Seeking Meaning→Positive Cognition→Intrinsic Motivation | Total Effect | 0.183*** | 0.136 | 0.230 | 0.024 | Partial mediation |
| | Direct Effect | 0.076*** | 0.034 | 0.118 | 0.021 | |
| | Indirect Effect | 0.107 | 0.075 | 0.143 | 0.017 | |
| Possessing Meaning→Positive Cognition→Intrinsic Motivation | Total Effect | 0.292*** | 0.254 | 0.330 | 0.019 | Partial mediation |
| | Direct Effect | 0.189*** | 0.148 | 0.229 | 0.021 | |
| | Indirect Effect | 0.104 | 0.070 | 0.141 | 0.018 | |

*$p<0.05$ ** $p<0.01$ *** $p<0.001$.

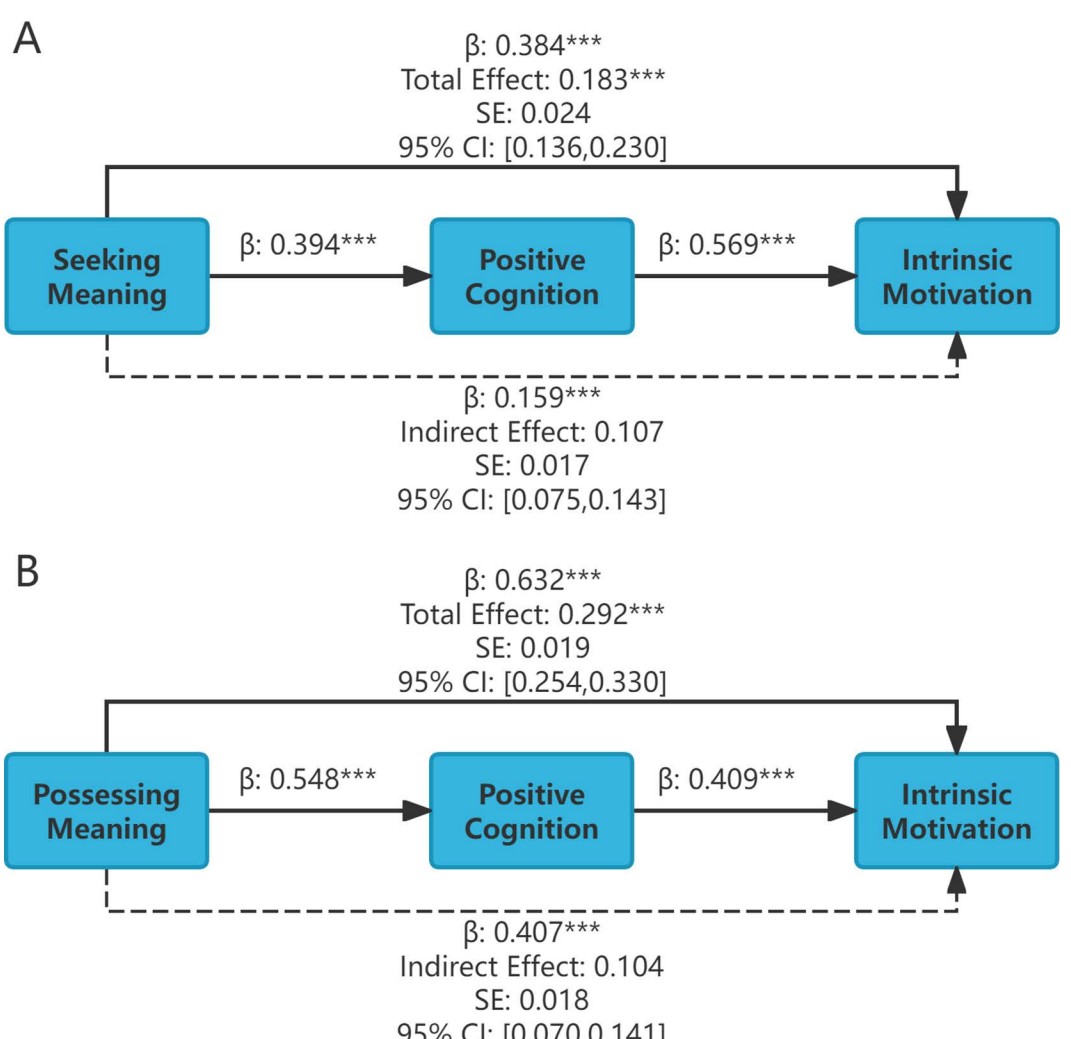

**Fig 3. Mediating effect pathway diagram between positive cognition, meaning in life, and intrinsic motivation.**

coefficient for the independent variable seeking meaning was 0.159, t = 3.553, p < 0.001, and for the independent variable with meaning, it was 0.407, t = 9.055, p < 0.001, validating the coefficient c', indicating that positive cognition plays a partial mediating role between the sense of meaning in life and intrinsic motivation.

Table 4 shows that the total effect value of the mediating path from seeking meaning to positive cognition to intrinsic motivation was 0.183 with a 95% confidence interval of [0.136, 0.230], which did not include zero, indicating that the total effect was valid. The indirect effect value was 0.107, with a 95% confidence interval of [0.075–0.143], which did not include 0, indicating that the mediating effect was valid. The direct effect value was 0.076, with a 95% confidence interval of [0.034–0.118], which did not include 0, indicating a partial mediating effect. The proportion of the mediating effect to the total effect was 0.107/0.183 × 100% = 58.5%, indicating that 58.5% of the impact of seeking meaning on intrinsic motivation was through positive cognition.

The total effect value of the mediating path from possessing meaning to positive cognition to intrinsic motivation was 0.292, with a 95% confidence interval of [0.254, 0.330], which did not include zero, indicating that the total effect was valid. The indirect effect value was 0.104, with a 95% confidence interval of [0.070–0.141], which did not include zero, indicating that the mediating effect was valid. The direct effect value was 0.189, with a 95% confidence interval of [0.148, 0.229], which did not include 0, indicating a partial mediating effect. The proportion of the mediating effect to the total effect was 0.104/0.292 × 100% = 35.6%, indicating that 35.6% of the impact of possessing meaning on intrinsic motivation was through positive cognition.

Mediating effects and path coefficient diagrams were drawn based on the results of the comprehensive mediating effects, as shown in S3 Fig.

**4.3.2. The mediating effect of positive cognition between sense of life meaning and extrinsic motivation.** Using SPSS 24.0 for hierarchical regression analysis and the bootstrap method to test the validity of the mediating effect, the results of the stepwise regression analysis are shown in Table 5, and the bootstrap mediating effect results are shown in Table 6.

From Table 5, it can be seen that in Step 1, the standardized regression coefficient of the independent variable seeking meaning is 0.303, t = 5.889, p < 0.001, and the standardized regression coefficient of the independent variable with meaning is 0.224, t = 4.258, p < 0.05, indicating that both seeking meaning and possessing meaning positively affect extrinsic motivation, thus verifying coefficient c. In step 2, the standardized regression coefficient of the independent

**Table 5. The mediating effect test of positive cognition between the sense of meaning in life and extrinsic motivation.**

| Step | Path | Unstandardized Coefficients | | Standardized Coefficients | t | p | Adjusted R² |
|---|---|---|---|---|---|---|---|
| | X→Y | B | SD | β | | | |
| 1 | Seeking Meaning→Extrinsic Motivation | 0.130*** | 0.022 | 0.303 | 5.889 | 0.000 | 0.089 |
| | Possessing Meaning→Extrinsic Motivation | 0.093*** | 0.022 | 0.224 | 4.258 | 0.000 | 0.047 |
| 2 | Seeking Meaning→Positive Cognition | 0.194*** | 0.024 | 0.394 | 7.946 | 0.000 | 0.153 |
| | Possessing Meaning→Positive Cognition | 0.262*** | 0.022 | 0.548 | 12.139 | 0.000 | 0.298 |
| 3 | Seeking Meaning→Extrinsic Motivation | 0.054* | 0.022 | 0.126 | 2.499 | 0.013 | 0.257 |
| | Positive Cognition→ Extrinsic Motivation | 0.390*** | 0.044 | 0.448 | 8.860 | 0.000 | |
| | Possessing Meaning→ Extrinsic Motivation | −0.029 | 0.023 | −0.070 | −1.249 | 0.104 | 0.247 |
| | Positive Cognition→ Extrinsic Motivation | 0.467*** | 0.049 | 0.536 | 9.584 | 0.000 | |

* p < 0.05 ** p < 0.01.

**Table 6. Bootstrap mediated effect test of positive cognition between the sense of meaning in life and extrinsic motivation.**

| Path | Effect | Effect Value | 95% CI | | SE | Conclusion |
|------|--------|-------------|--------|--------|-----|-----------|
| | | | LLCI | ULCI | | |
| Seeking Meaning→Positive Cognition→Extrinsic Motivation | Total Effect | 0.130*** | 0.087 | 0.173 | 0.022 | Partial mediation |
| | Direct Effect | 0.054* | 0.012 | 0.097 | 0.022 | |
| | Indirect Effect | 0.076 | 0.048 | 0.110 | 0.016 | |
| Possessing Meaning→Positive Cognition→Extrinsic Motivation | Total Effect | 0.093*** | 0.050 | 0.136 | 0.022 | Complete mediation |
| | Direct Effect | −0.029 | −0.075 | 0.017 | 0.023 | |
| | Indirect Effect | 0.122 | 0.090 | 0.159 | 0.017 | |

$* p<0.05 ** p<0.01 *** p<0.001.$

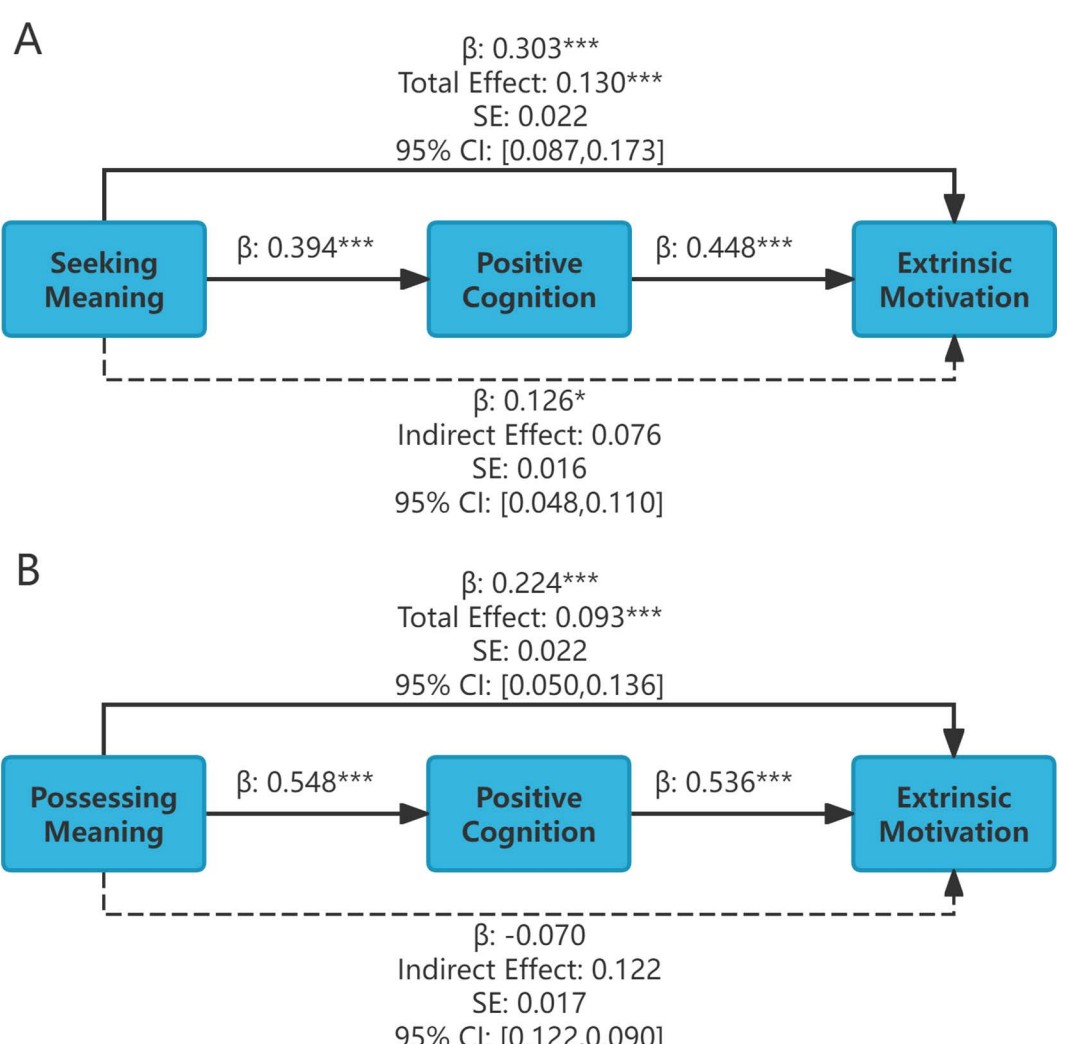

**Fig 4. Mediating effect pathway diagram between positive cognition, meaning in life, and extrinsic motivation.**

variable seeking meaning was 0.394, t = 7.946, p < 0.001, and the standardized regression coefficient of the independent variable with meaning was 0.548, t = 12.139, p < 0.001, indicating that both seeking meaning and possessing meaning positively affected positive cognition, thus verifying Coefficient a. On Tep 3, when the independent variable was seeking meaning, the standardized regression coefficient for the mediating variable, positive cognition, was 0.448 (t = 8.860, p < 0.001). When the independent variable possessed meaning, the standardized regression coefficient for the mediating variable, positive cognition, was 0.536 (t = 9.584, p < 0.001), indicating that positive cognition positively affects extrinsic motivation. This verifies coefficient b, indicating that positive cognition plays a mediating role between sense of life meaning and extrinsic motivation. The standardized regression coefficient of the independent variable seeking meaning was 0.126 (t = 2.499, p < 0.05), indicating that positive cognition partially mediated the relationship between seeking meaning and extrinsic motivation. The standardized regression coefficient of the independent variable with meaning was −0.070, t = −1.249, p > 0.05, not verifying the coefficient c', indicating that positive cognition plays a complete mediating role between possessing meaning and extrinsic motivation.

From Table 6, it can be seen that the total effect value of the mediating path from seeking meaning to positive cognition to extrinsic motivation is 0.130, with a 95% confidence interval of [0.087, 0.173], which does not include 0, indicating that the total effect is valid. The indirect effect value was 0.076, with a 95% confidence interval of [0.048, 0.110], which did not include zero, indicating that the mediating effect was valid. The direct effect value was 0.054, with a 95% confidence interval of [0.012–0.097], which did not include 0, indicating a partial mediating effect. The proportion of the mediating effect to the total effect was 0.076/0.130 × 100% = 58.5%, indicating that 58.5% of the impact of seeking meaning on extrinsic motivation occurred through positive cognition.

The total effect value of the mediating path from meaning possession to positive cognition to extrinsic motivation was 0.093, with a 95% confidence interval of [0.050, 0.136], which did not include zero, indicating that the total effect was valid. The indirect effect value was 0.122, with a 95% confidence interval of [0.090,0.159], which did not include 0, indicating that the mediating effect was valid. The direct effect value was −0.029, with a 95% confidence interval of [−0.075, 0.017], which included 0, indicating a complete mediating effect. This finding indicates that the effect of possessing meaning on extrinsic motivation occurs primarily through positive cognition.

Mediating effects and path coefficient diagrams were drawn based on the results of the comprehensive mediating effects, as shown in S4 Fig.

## 5. Discussion

This study investigated the interrelationships among meaning in life, positive cognition, and learning motivation in a sample of Chinese university students, with a primary focus on the mediating role of positive cognition. Our findings provide a more nuanced understanding of the psychological pathway linking a student's sense of meaning to their academic motivation, highlighting positive cognition as a key psychological mechanism.

### 5.1. The interconnectedness of meaning, cognition, and motivation

Before testing the mediation model, we first established the foundational relationships between the core variables. Consistent with our hypotheses and previous research [1,4], our results confirmed significant positive correlations between meaning in life (both presence and search), positive cognition, and learning motivation (both intrinsic and extrinsic). Students reporting higher levels of meaning in life also tended to report more positive cognitive styles and greater learning motivation. Similarly, a strong positive association was found between positive cognition and learning motivation. These significant bivariate relationships provided the necessary groundwork for proceeding with the mediation analysis, suggesting that these constructs are meaningfully interlinked and a pathway model is plausible.

### 5.2. The core mechanism: How positive cognition translates meaning into motivation

The central contribution of this study is the elucidation of the mediating role of positive cognition. Our findings suggest that the association between meaning in life and learning motivation is not just direct, but is significantly channeled through an individual's cognitive style.

Regarding intrinsic motivation, our analysis revealed that positive cognition partially mediated the relationship for both presence of meaning and search for meaning. This suggests a two-pronged pathway. There appears to be a direct association between a sense of meaning and a student's inherent enjoyment of learning, which aligns with Self-Determination Theory's emphasis on purpose fuelling autonomous motivation [9,29]. However, a substantial portion of this relationship operates indirectly. A strong sense of meaning—whether currently held or actively sought—is associated with a more optimistic and constructive cognitive orientation. In line with the Broaden-and-Build Theory [13], this positive cognition may then broaden a student's perspective on academic tasks, helping them see challenges as opportunities for growth, which in turn nurtures their intrinsic passion for learning.

The most compelling insights emerged from the notably distinct mediation patterns for extrinsic motivation. The association between presence of meaning and extrinsic motivation was fully mediated by positive cognition. This is a critical finding. It suggests that for students who already feel their lives are meaningful, this sense of purpose is not directly associated with the pursuit of external rewards (like grades). Instead, its relationship with such goals appears to be entirely contingent upon a positive cognitive framework. Essentially, their stable sense of meaning is linked to the drive for external achievement only when it is accompanied by optimistic expectations and positive self-appraisals.

Conversely, the link between the search for meaning and extrinsic motivation was partially mediated. This implies that students actively searching for meaning may have both a direct and an indirect link to pursuing external goals. The direct link might reflect a belief that academic achievements or a successful career are instrumental in their quest to find purpose. The indirect link, operating through positive cognition, mirrors the other findings: the process of searching is associated with positive thinking, which in turn is related to motivation for external outcomes.

In sum, these differentiated pathways highlight positive cognition not merely as a correlate, but as a central psychological mechanism. It appears to function as a "psychological converter," as we proposed. Drawing on our theoretical framework, a sense of meaning may satisfy fundamental needs for autonomy and purpose (SDT), which is associated with a more positive cognitive style. This cognitive style then builds the psychological resources (Broaden-and-Build) necessary to sustain motivation for both intrinsic and extrinsic academic pursuits.

## 6. Conclusions

This study investigated the psychological pathway connecting meaning in life to learning motivation among Chinese university students, positioning positive cognition as a key mediating mechanism. The findings confirm our central hypotheses and offer a more nuanced understanding of this relationship, providing valuable insights for both theory and practice.

### 6.1. Major findings

The study yielded three primary findings. First, we confirmed significant positive correlations between meaning in life (both the presence of and search for meaning), positive cognition, and learning motivation (both intrinsic and extrinsic). Students with a stronger sense of life meaning reported more positive cognitive styles and higher academic motivation, establishing the foundational links for our model.

Second, and most importantly, our analysis demonstrated that positive cognition serves as a crucial mediator in the relationship between meaning in life and learning motivation. It functions as a "psychological converter," translating an abstract sense of purpose into tangible academic drive.

Third, we uncovered distinct mediating pathways depending on the type of motivation. For intrinsic motivation, positive cognition acted as a partial mediator for both the presence of and search for meaning. This suggests that while a sense of purpose has a direct link to a student's innate love of learning, a significant portion of that effect is channeled through a positive mindset. For extrinsic motivation, the pathways were even more distinct: the relationship between the presence of meaning and extrinsic motivation was fully mediated by positive cognition, indicating that a student's existing sense of purpose influences their drive for external rewards entirely through the lens of positive thinking. Conversely, the link between the search for meaning and extrinsic motivation was only partially mediated, suggesting the search itself may directly fuel a desire for external validation.

## 6.2. Implications

These findings carry significant theoretical and practical implications. Theoretically, this study contributes to the literature by integrating Self-Determination Theory and the Broaden-and-Build Theory to explain the cognitive mechanisms underlying motivation. By demonstrating that positive cognition is a key bridge between an existential state (meaning in life) and a behavioral outcome (learning motivation), our model moves beyond simple correlation to offer a more nuanced, process-oriented explanation.

Practically, the results provide actionable guidance for universities and educators. The finding that positive cognition is a powerful mediator suggests that interventions should not only focus on helping students explore "the why" of their lives but also on teaching them "the how" of constructive thinking. Educational institutions could develop and implement programs aimed at cultivating positive cognitive skills, such as workshops on cognitive reframing, optimism, and gratitude. Such initiatives could be a highly effective strategy for boosting students' academic engagement, enhancing their motivation, and ultimately supporting their overall psychological well-being in a competitive academic environment.

## 6.3. Limitations and recommendations for future studies

Despite its contributions, this study has several limitations that open avenues for future research. First, the cross-sectional design precludes the ability to establish causal relationships or track developmental changes over time. Future research should employ longitudinal or experimental designs, perhaps by assessing students before and after a meaning-centered or positive psychology intervention, to better understand the temporal dynamics between these variables.

Second, our sample was drawn from a single university in Chongqing, which may limit the generalizability of the findings. To enhance external validity, future studies should recruit more diverse samples from various geographical regions, types of institutions (e.g., public vs. private, research-focused vs. teaching-focused), and academic disciplines. Extending this research to other age groups, such as adolescents or early-career professionals, would also provide a more complete picture of these psychological dynamics across the lifespan.

Third, a key limitation is the low average variance extracted (AVE = 0.28) for the extrinsic motivation subscale, even with adequate composite reliability (CR = 0.72) and established discriminant validity (via Fornell-Larcker criterion and HTMT < 0.85) (Specific data on HTMT are not presented here but are available from the corresponding author upon reasonable request) [37–39]. This suggests items incompletely capture the construct's variance, which may moderately weaken the interpretability of its construct validity. Future work should refine the scale by expanding items to better encompass extrinsic motivation's sub-dimensions (e.g., external regulation) and applying stricter selection thresholds (e.g., higher factor loadings).

Finally, Operationalizing Positive Cognition through the single subscale of positive attention is grounded in theoretical and empirical support: positive attention functions as a foundational mechanism in positive cognitive processes [13,40], correlates with broader tendencies such as positive reappraisal and stress resilience [41,42], and stands as a reliable, measurable indicator validated in applied research [43–45]. However, a limitation of this approach is that focusing solely on positive attention may overlook other facets of Positive Cognition, including positive interpretation and future-oriented

expectations. Future studies are thus encouraged to adopt comprehensive measures like the Positive Cognitive Style Questionnaire to more fully capture the construct's multidimensionality Figs 1–4.

## Supporting information

**S1 Appendix. Data collection https://doi.org/10.6084/m9.figshare.29504990.(xlsx).**
(XLSX)

**S2 Appendix. Meaning in life questionnaire (MLQ) Positive cognitive questionnaire Learning motivation questionnaire.**
(PDF)

**S1 Fig. Theoretical model diagram.**
(TIF)

**S2 Fig. The hypothesized mediation model.**
(TIF)

**S3 Fig. Mediating effect pathway diagram between positive cognition, meaning in life, and intrinsic motivation.**
(TIF)

**S4 Fig. Mediating effect pathway diagram between positive cognition, meaning in life, and extrinsic motivation.**
(TIF)

## Acknowledgments

The authors thank two anonymous reviewers for their helpful comments on previous versions of this paper.

## Author contributions

**Conceptualization:** Shi-hong Liu, Xiao-li Gong, Lei-ying Xiang.

**Data curation:** Xiao-li Gong, Ming-hui Kong.

**Formal analysis:** Xiao-li Gong, Ming-hui Kong.

**Investigation:** Qi Ji, Qiao-qiao Li.

**Methodology:** Qi Ji, Qiao-qiao Li.

**Project administration:** Zhi-ru Zhu.

**Resources:** Zhi-ru Zhu.

**Software:** Xiao-hui Du.

**Supervision:** Xiao-hui Du, Lei-ying Xiang.

**Visualization:** Zhi-ru Zhu.

**Writing – original draft:** Shi-hong Liu, Xiao-li Gong.

**Writing – review & editing:** Shi-hong Liu.

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
