## [Decision Letter · Decision Letter 0]

16 Jun 2025

PONE-D-25-25397The Mediating Role of Positive Cognition in Connecting Life Meaning and Learning Motivation: A Cross-Sectional Study of Chinese College StudentsPLOS ONE

Dear Dr. Zhu,

Thank you for submitting your manuscript to PLOS ONE. After careful consideration, we feel that it has merit but does not fully meet PLOS ONE’s publication criteria as it currently stands. Therefore, we invite you to submit a revised version of the manuscript that addresses the points raised during the review process.

We look forward to receiving your revised manuscript.

Kind regards,

Henri Tilga, PhD

Academic Editor

PLOS ONE

Journal Requirements:

3. Thank you for uploading your study's underlying data set. Unfortunately, the repository you have noted in your Data Availability statement does not qualify as an acceptable data repository according to PLOS's standards.

4. Please amend the manuscript submission data (via Edit Submission) to include author Zhiru Zhu.

5. Please amend your authorship list in your manuscript file to include author Zhi-ru Zhu.

6. Please remove your figures from within your manuscript file, leaving only the individual TIFF/EPS image files, uploaded separately. These will be automatically included in the reviewers’ PDF.

Reviewers' comments:

Reviewer's Responses to Questions

**Comments to the Author**

1. Is the manuscript technically sound, and do the data support the conclusions?

Reviewer #1: Partly

Reviewer #2: Yes

2. Has the statistical analysis been performed appropriately and rigorously? 

Reviewer #1: Yes

Reviewer #2: Yes

3. Have the authors made all data underlying the findings in their manuscript fully available?

Reviewer #1: Yes

Reviewer #2: No

4. Is the manuscript presented in an intelligible fashion and written in standard English?

Reviewer #1: No

Reviewer #2: Yes

5. Review Comments to the Author

Reviewer #1: The topic is interesting and meaningful. I think you can have an great improvement in the quality of the paper. Here are major poings for you and please see the details in the attached file.

1. Introduction is too long. There should be a new part "Literature Review"

2. Theoretical fundations (SDT) should be highlighted in a section in Literature Review.

3. The gap or the need for the study should be rewritten. The present version is not well developed in saying this point.

4. The instruments should be added some descriptions of validity and reliablity.

5. The findings should be given more space for mediation roles report. Correspondingly, the discussion part should say more about this report.

Reviewer #2: This study explores the mediating role of positive cognition in the relationship between life meaning and learning motivation among Chinese university students. The topic is timely and relevant, particularly within the context of growing interest in positive psychology, student well-being, and motivation research. The manuscript is generally well-written and organized, with a logical flow from introduction to discussion. However, several critical issues must be addressed to improve the manuscript's scientific rigor, clarity, and contribution to the field.

1. The conceptual rationale for using positive cognition as the sole mediator is underdeveloped. Why was positive cognition chosen over other potentially relevant mediators such as self-efficacy, resilience, or optimism? The manuscript lacks a detailed theoretical model illustrating the hypothesized relationships (figures). Inclusion of a theoretical model earlier would enhance clarity. I suggest to expand the theoretical justification for the choice of mediator. Consider integrating a broader theoretical lens (e.g., Self-Determination Theory, Broaden-and-Build Theory) to justify the mechanisms. The following article may help how to structure article and report methods. Du, M. & Ashraf, M. A. (2025). Antecedents of environmental protection participation intentions among Chinese adolescents: The perspectives of rational choice, moral norms, and supporting environment. British Educational Research Journal, 00, 1–25. https://doi.org/10.1002/berj.4168

2. The Learning Motivation Questionnaire’s extrinsic motivation subscale shows poor construct validity (AVE = 0.28). This raises concerns about the adequacy of this scale for capturing extrinsic motivation. In addition, Positive Cognition is operationalized through a single subscale (positive attention), which might not fully represent the broader construct of positive cognition.

3. Although bootstrapping was used, no justification was provided for the sample size’s adequacy in mediation analysis. Hierarchical regression analysis is appropriate, but more sophisticated mediation analysis using SEM or PROCESS macro would provide more robust insights.

4. The discussion sometimes overstates the findings, particularly regarding causality (e.g., phrases like “enhancing life meaning fosters...” imply causation, which is inappropriate for cross-sectional data).

The authors mention that 58.5% of the effect is mediated but fail to clarify whether this is a proportion of the standardized total effect or unstandardized. Figure 1 and Figure 2 are insufficiently detailed (e.g., missing p-values, standard errors).

5. Revise the manuscript for conciseness and consistency in terminology. Consider reorganizing or trimming redundant sections (especially in the Discussion and Limitations).

6. Several citations are from lesser-known journals or in Chinese; efforts should be made to include more international, high-impact sources if possible. The literature review is limited in its scope and in-depth analysis of the topic. Massive literature is available on meaning and learning motivation in Chinese and other nation's context.

Feng, W., Wu, P., Lv, S. et al. The relationship between meaning in life and self-regulated learning among college students: the mediating effect of psychological capital and the moderating effect of phubbing. BMC Psychol 13, 566 (2025).

Cheng, L., Chen, Q., & Zhang, F. (2021). Mediating effects of meaning in life on the relationship between general self-efficacy and nursing professional commitment in nursing students: A structural equation modeling approach. Medicine, 100(29), e26754. https://doi.org/10.1097/MD.0000000000026754

7. Why is Glossary added to the article? There is no need of glossary in academic journal article.

8. The abstract is too dense and technical; simplify and clarify mediation results for a broader audience.

9. Provide the questionnaire and other necessary data in supplementary materials.

6. PLOS authors have the option to publish the peer review history of their article (what does this mean? ). If published, this will include your full peer review and any attached files.

**Do you want your identity to be public for this peer review?** For information about this choice, including consent withdrawal, please see our Privacy Policy .

Reviewer #1: No

Reviewer #2: No

---

## [Author Response · Author response to Decision Letter 1]

12 Jul 2025

Our detailed point-by-point response is provided in the uploaded file named "Response to Reviewers".

---

## [Decision Letter · Decision Letter 1]

1 Aug 2025

Meaning in life, positive cognition, and learning motivation: A mediational analysis among Chinese college students

PONE-D-25-25397R1

Dear Dr. Zhu,

We’re pleased to inform you that your manuscript has been judged scientifically suitable for publication and will be formally accepted for publication once it meets all outstanding technical requirements.

Kind regards,

Henri Tilga, PhD

Academic Editor

PLOS ONE

Additional Editor Comments (optional):

Reviewers' comments:

Reviewer's Responses to Questions

**Comments to the Author**

1. If the authors have adequately addressed your comments raised in a previous round of review and you feel that this manuscript is now acceptable for publication, you may indicate that here to bypass the “Comments to the Author” section, enter your conflict of interest statement in the “Confidential to Editor” section, and submit your "Accept" recommendation.

Reviewer #1: All comments have been addressed

2. Is the manuscript technically sound, and do the data support the conclusions?

Reviewer #1: Yes

3. Has the statistical analysis been performed appropriately and rigorously? 

Reviewer #1: Yes

4. Have the authors made all data underlying the findings in their manuscript fully available?

Reviewer #1: Yes

5. Is the manuscript presented in an intelligible fashion and written in standard English?

Reviewer #1: Yes

6. Review Comments to the Author

Reviewer #1: thank you so much for this professional revision version, which has addressed all my concerns. I recommend it to be published.

7. PLOS authors have the option to publish the peer review history of their article (what does this mean? ). If published, this will include your full peer review and any attached files.

**Do you want your identity to be public for this peer review?** For information about this choice, including consent withdrawal, please see our Privacy Policy .

Reviewer #1: No

---

## [Editor Report · Acceptance letter]

PONE-D-25-25397R1

PLOS ONE

Dear Dr. Zhu,

I'm pleased to inform you that your manuscript has been deemed suitable for publication in PLOS ONE. Congratulations! Your manuscript is now being handed over to our production team.

Kind regards,

on behalf of

Dr. Henri Tilga

Academic Editor

PLOS ONE